# Longitudinal employment patterns and parental health: A cross-country look

Wen-Jui Han[1]*, Johanna Carrasco Saravia[2], Matthias Pollmann-Schult[3], Tinh Doan[4], Jianghong Li[2,5]

1 New York University, New York, New York, United States of America, 2 WZB Berlin Social Science Center, Berlin, Germany, 3 Siegen University, Siegen, Germany, 4 SA Centre for Economic Studies, School of Economics, Adelaide University, Adelaide, Australia, 5 Curtin Medical School, Curtin University Perth, Perth, Western Australia, Australia

* wjh3@nyu.edu

## Abstract

### Study aims

Using a cross-country lens, we investigate the links between longitudinal work trajectories and health among parents with children under age 18.

### Background

Employment serves as a valuable resource, affording us a decent standard of living. The rising dominance of digital and technology, together with the service economy since the 1980s, has transformed the utility of employment from a resource to a vulnerability, subjecting more families to uncertain, unstable, and insecure work. Nonstandard work schedules or shiftwork, which often fall outside regular 9-to-5 daytime hours and can be unpredictable, carry potential health consequences.

### Methods

Using the longitudinal data from Australia (HILDA), Germany (SOEP), the UK (UKHLS), and the US (NLSY79), we used sequence analysis to first chart parental work schedule patterns between three stages of the life course, 25–34, 35–44, and 45–54, to show the changes and transitions in work patterns. We then conducted multivariate regression analysis to examine how variations in parental work patterns may shape individual health (i.e., physical and mental health) at ages 35/40, 45/50, and 55/60 while controlling for a rich set of sociodemographic characteristics.

### Results

Our sequence analyses uncovered roughly 4–6 work patterns during those three periods, revealing the heterogeneities of parental work trajectories that might correspond to childrearing demands and their sociodemographic backgrounds. We also found that mainly not-working pattern or volatile work arrangements (e.g., switching

**Data availability statement:** All relevant data are within the paper and its Supporting information Files (S1 File).

**Funding:** The author(s) received no specific funding for this work.

**Competing interests:** The authors have declared that no competing interests exist.

between daytime and non-daytime hours) were associated with significantly poorer physical and mental health; however, the persistence and magnitude of these associations varied by country.

## Conclusions

This study advances our understanding of the critical role of employment in our health from a cross-country perspective and bears important implications for the intergenerational transmission of employment and health vulnerabilities.

## Introduction

Universally, parents invest both time and money to strive for a decent standard of living and quality family life for the benefit of the next generation. Employment provides the means and resources to afford a decent living standard, but might come at the cost of not spending enough (quality) time with children. Furthermore, the increasing volatility in work schedules since the 1980s has made some work arrangements more vulnerable to health and well-being than others. Specifically, the globalization and technological advances coupled with service economies have diversified and polarized the labor market with an increasing share of workers required to work outside traditional 9–5 daytime hours or so-called nonstandard schedules or shift work [1–3]. In developed economies, 20–40% of the workforce works evenings, nights, rotating or split shifts, irregular hours, or on weekends [4–7]. These nonstandard schedules have been referred to as "unsociable" to family routines and social activities [8], and have been shown to carry a physical and mental health toll on workers [9,10], due to the likely feature of low wage (thus low income) and volatility in hours and scheduling [11,12]. Yet, we still lack a comprehensive understanding of parental well-being in the context of shift work. Parental well-being is not only important in its own right but also critical to child well-being, with implications for intergenerational transmission of health disparities. In this paper, we use the terms "nonstandard work schedules" and "shift work" interchangeably, as global literature has used either term to define work arrangements that are atypical daytime hours, such as early mornings, evenings, nights, irregular or unpredictable hours.

The global prevalence of parental nonstandard work schedules warrants careful examination of parental health for at least two reasons. First, a large body of scholarship has shown that holding jobs with nonstandard work schedules has adverse health consequences, primarily due to unstable, unpredictable, and insecure natures [9,13,14]. Poor health incurs economic and productive costs that are not only borne by the employers but also by society as a whole [15–17] with an estimated annual health-related productivity losses costing employers $530 billion [15], which are largely due to medical care expenses, worker compensation, and time delays and loss due to worker absence. Second, a recent review of empirical evidence (mostly on the US) found that low wages and low family income, a persistent feature of jobs with nonstandard schedules, perpetuate the cycle of economic disadvantage [18].

These disadvantages limit families' ability to provide adequate nutrition and housing, access to medical care, and educational opportunities for their children, thereby jeopardizing their children's health and well-being [18]. However, the present empirical evidence has been inconclusive at best regarding the links between work schedules and parental health, partly due to methodological differences (e.g., cross-sectional vs. longitudinal; nationally representative data vs. otherwise) and country contexts [3,7,19].

Building on the social determinants of health (SDOH) and a life-course perspective along with the extant scholarship on the links between shift work and health, this study examines the extent to which various work arrangements related to shift work are associated with parental health, using a life course lens in a cross-country comparison (Australia, Germany, the UK, and the US). We pose the following four research questions: (1) How do parental work patterns evolve over the life course? (2) Are there cross-country differences in parental work patterns? (3) Are certain work patterns associated with worse health outcomes than others? And (4) Do the strengths of these associations vary across countries? We use a cross-country lens to recognize the global phenomenon of nonstandard work schedules and to determine whether their health consequences might be felt universally among parents, despite inherent heterogeneity across countries. We acknowledge the inherent heterogeneity across countries by using an institutional lens to shed light on the similarities and differences in the health consequences of nonstandard work schedules among parents, in addition to our methodology (detailed in the Methods section) addressing some heterogeneity issues in the data (e.g., data harmonization). Specifically, building upon scholarship attesting the critical roles of labor market and social protection provisions (e.g., working time regulation, collective bargaining, childcare policies and services) in shaping the prevalence of shift work [20], we anticipate these institutional provisions playing an important role to understand the potential similarities and differences in parental health due to work schedules, thereby shedding informative light on the inconclusive empirical evidence from the extant research.

Of importance, establishing causality is an undeniably powerful tool for informing policies and practices to address how and why work schedules might shape parental health. While the use of longitudinal data does not establish causality due to its non-experimental nature (and unethical for an experimental study in our case of parental work schedules), our life-course approach, achieved by sequence analysis, provides a strong tool to carefully delineate the work pathways over time to unfold the dynamics in parents juggling work arrangements and hence their later health. For one thing, the longitudinal approach has the advantage of reducing measurement noise by capturing long-term patterns more accurately than cross-sectional data, for example, by distinguishing between individuals who repeatedly reported nonstandard work schedules over the years and those who worked such a schedule only a few times in 10 years. For the other, sequence analysis allows us to establish the proper temporal sequence between exposure and outcome, a necessary prerequisite, although not a sufficient condition, for making causal inference. Our findings, therefore, contribute to the accumulation of empirical evidence using longitudinal data and appropriate statistical tools, taking a step toward establishing causality when experimental designs are impossible (e.g., smoking and cancer).

### The context

Since the 1980s, globalization and technological advances have transformed and diversified the global labor market to serve a 24−7 economy around the globe by having an increasing share of highly skilled and high-paying jobs (e.g., engineers, computer scientists) and low-wage jobs (e.g., cashiers, health aids, truck drivers, food preparation) that oftentimes require nonstandard work schedules [2].

The prevalence of unsociable work schedules (e.g., evenings, nights, weekends) and unpredictable work time (rotating, irregular, or on-call) is substantial. In Europe, 17% of individuals frequently work in the evening or at night and 34% regularly work on weekends [5]. In Australia, 19% of all workers typically work nonstandard schedules, and almost 50% work on weekends, as of 2023 [4]. In the US, nearly 20 percent of the workforce work in the evenings, nights, or weekends [6,21], with higher prevalence among some groups (23% for African Americans, 27% for part-time workers, 36%

in the service sector [6]). The prevalence is even higher among low-wage workers in the United States, with estimates suggesting that 28% and 50% of low-wage workers have nonstandard work schedules [22]. Parents are no exception to the prevalence of shift work. For example, in Europe, on average, 34% of parents work such schedules and this prevalence rises to 42% for southern European and Central Eastern European countries [23]. In Australia, the figure is as high as 60% among parents with dependent children [24]. In the UK, nearly 25% of employed mothers and 35% of employed fathers work nonstandard hours [7]. In the US, 36% of fathers and 40% of mothers work nonstandard schedules during their prime working ages (i.e., 22–49) [25].

## Nonstandard work schedules as a social determinant of health throughout the life course and across countries

We draw upon social determinants of health (SDOH) scholarship [26] to investigate the critical links between parental work schedules and health. SDOH includes the broader social, economic, and policy environments, such as employment conditions, social support, and access to resources, that have greater influence on health and well-being than genetic influences [27]. Hence, SDOH may include, but are not limited to, social gradient (e.g., gender, race, ethnicity), early-life experiences, social exclusion, employment, working conditions, social support, food insecurity, and access to services (e.g., healthcare). Critical links between SDOH and health have been proven by the strong empirical evidence that individuals with limited access to resources such as education, social protection, and job opportunities have a higher risk of illness and death, due to limited access to health-promotion resources [27]. Nonstandard work schedules are considered as a SDOH for its hours (e.g., evenings or nights) tending to be out of sync of biological rhythms that comprises our sleep routines and sleep duration and quality, thus weakening our immunity and elevating the ristk for a range of diseases [28]; its uncertain, unstable, or unpredictable hours (e.g., irregular or rotating shifts or involuntary schedule changes) and tendency of shifting between working and joblessness (insecure employment) that may bring chaos and stress to daily life; and its likelihood of being part-time hours that may bring insufficient economic resources [2,20,29]. Nonstandard work schedules may also serve as a manifestation of structural inequality as workers with relatively disadvantaged characteristics (e.g., lowly educated, racial-ethnic minority group) tend to be subject to jobs requiring nonstandard schedules [2,20,25,29]. Thus, work not only serves as a resource but also a source of vulnerability, shaping life trajectories that produce inequality in accessing work-related resources and achieving social status, thereby creating and reinforcing health disparities.

We also draw on the life course perspective [30] because SDOH perspective asserts a lifelong process by which our living and working conditions acutely and cumulatively influence our health throughout our lives. Specifically, a life course perspective considers that health and well-being are lifelong products of cumulative experiences shaped by our social and economic conditions. Hence, a life course approach allows us to capture the complexities of dynamic changes and transitions in work life, consequently illuminating how engaging in nonstandard work schedules and transitioning between different work arrangements throughout our lifetimes, as a cumulative exposure, may shape later health and well-being. We explicitly build upon the principle that the past shapes the future, taking advantage of longitudinal data, to directly examine the relationship between one's earlier work experience and subsequent health outcomes at various stages of adulthood (e.g., 35/40, 45/50, 55/60). We posit that the timing and sequencing of transitions may differentially affect one's later health and explore this hypothesis in our life course analysis.

Another essential tenet of SDOH is the critical role of the social, economic, political, and policy environments in which one resides. Indeed, SDOH emphasizes that our living conditions, shaped by macro-environments including policies, play a critical role in determining who has access to essential resources that promote and maintain our health and well-being. Two people with the same characteristics (e.g., gender, race, education, employment) may have drastically different health outcomes, as one might reside in a society that provides universal healthcare affordable to everyone, while the other resides in a society heavily reliant on private healthcare unaffordable and unreachable to many [27]. We adopt a cross-country lens to explore the idea that part of SDOH involves understanding how and why people experience various

living conditions. While we believe parents share a universal goal of nurturing the next generation by investing in both time and money, the context (e.g., employment and family policies) might drive parental choices of various work arrangements, and the differences in national contexts might also drive or modify the associations between work arrangements and parental health, with implications for the intergenerational health disparity.

The four countries examined in this study differ significantly in key institutional factors that may influence the relationship between nonstandard work schedules and parental health, particularly regarding childcare provision and working time regulations. For example, childcare costs differ substantially across the four countries [31]. Germany has comparatively low out-of-pocket childcare costs due to extensive public subsidies and regulated fees. In contrast, childcare is considerably more expensive in Australia and the US, where provision relies more heavily on market-based systems and parental contributions. The UK occupies an intermediate position: although public support reduces costs for some families, childcare fees remain high by international standards [20,31]. On average, families with two children in Germany spend only about 1% of earnings on childcare, compared with substantially higher shares in the US (32%), the UK (49%), and Australia (60%) [31]. Furthermore, Germany and the UK offer generous public childcare services: in 2017, approximately 95% of children aged 3–5 were enrolled in public childcare in both countries; wherease the enrollment rates were considerably lower in Australia (84%) and the US (66%) [31]. Parents who lack access to formal or affordable childcare may therefore resort to working evenings, nights, or weekends as a tag-team parenting to accommodate childcare needs so that one parent is home to reduce childcare costs [32,33]. Consistent with this pattern, cross-national research shows that parents residing in countries with lower levels of childcare provision are more likely to work evening or night hours than those living in countries with more extensive childcare systems, suggesting that limited access to formal childcare increases parents' reliance on nonstandard work schedules [34].

However, social protection policies, such as childcare, are only part of the story. Structural and labor market institutions play another equally important, if not more important, role in shaping parental employment and work behaviors, such as work time regulation [20]. For example, the German government, like those of other Western European countries, tends to be more involved in regulating the labor market and employment policies, including unionization and collective bargaining practices [20]. These labor market regulations help ensure decent wages and protective hours, which, in turn, may cushion the potential health harms of nonstandard work schedules. In contrast, Australia, the UK, and the US are characterized by low government involvement, driven and reinforced by strong ideologies that endorse free markets and individualism [20]. The consequences are low union density, and employees have little collective bargaining power and few institutionalized employment guarantees (e.g., job security). As a result, these labor market provisions may exacerbate the adverse links between nonstandard work schedules and health. However, when it comes to working time flexibility, Australia ranks the highest for parents have the legal right to request flexible working hours and the option to work from home, followed by Germany and the UK, while the US offers the least flexibility. For example, irregular or variable work schedules are common in the US, particularly in sectors like retail and food service [2], limiting workers' control over their schedules. Taken together, these patterns reveal that Germany combines generous and affordable public childcare with relatively high working time flexibility. The US, by contrast, offers limited public childcare and low flexibility, exacerbated by a fragmented health care system that disproportionately disadvantages workers in low-wage or unstable jobs. Australia offers high working time flexibility but limited access to childcare, while the UK stands out for its generous public childcare but relatively low flexibility in working hours, compared to Germany; however, in both countries, employer discretion remains substantial, particularly in sectors reliant on casual or temporary arrangements. These institutional differences shape both the prevalence of nonstandard work schedules and the extent to which the health consequences of various work trajectories.

## Health consequences of nonstandard work schedules

Global scholarship has documented adverse physical and mental health for workers with nonstandard work schedules, with some work arrangements (e.g., night hours) posing greater risks than others [9–11,13,25,35–38]. For example, not

working or experiencing joblessness may result in the loss of essential family income, leading to poor economic well-being and inadequate access to critical resources, which in turn can contribute to poorer physical and mental health [9,25,35]. The increasingly widespread practice of requiring working during early morning, evening, night, or irregular hours has also challenged workers' well-being, with negative consequences ranging from sleep routines misaligned with biological rhythms (due to night shifts or irregular shifts) to a rise in health behaviors such as smoking, drinking, poor diet, or lack of physical exercise [9,11,38–40]. These health behaviors can lead to adverse physical health consequences, including both chronic conditions such as type 2 diabetes and heart disease [11,40,41], poor mental health such as distress, anxiety, and depression [11,40,42,43], and poor self-assessed subjective health [44,45]. Engaging in nonstandard work schedules has also been shown to disrupt family routines, such as not being able to spend (quality) time together during evening and nighttime hours or not eating dinner as a family [46]. Notably, empirical evidence shows that unpredictable work schedules (e.g., rotating, on-call shifts) are particularly disruptive to workers and their families, compared to fixed evening or night shifts [12,47]. Studies have also documented challenges in securing stable and quality childcare for children when parents have nonstandard work schedules [48,49]. Moreover, parents are often compelled to use multiple arrangements including sibling care or leaving children unsupervised to accommodate unpredictable and unstable schedules [50]. As a result, nonstandard work schedules have been found to be associated with poorer academic and socioemotional well-being in children (see the review in [51]). Nonstandard work schedules can thus take a physical, psychological, and social toll on workers and their family members, carrying an intergenerational inequality.

However, people may voluntarily choose specific work schedules for various reasons during different life stages and may be forced to work particular schedules at other times. For example, young workers without seniority might be required to work nonstandard schedules. Workers in their 30s may have young children and intentionally opt for schedules that allow them to be with their children during the day [32,33]. They may also coordinate with partners to maximize their time with children and reduce childcare expenses [49,52]. The variations are critical because when one voluntarily chooses a nonstandard schedule, the flexibility and emotional fulfillment gained may outweigh the physical and mental toll of working such schedules [52]. Today, parents are increasingly working long, nonstandard, or unpredictable hours [7,22–25]. Hence, some parents may opt for specific combinations of work schedules to support each other, while other parents may do so to reduce financial burdens [32,33,52].

We expect that the association between parental work schedules and poor health would be similar to that for workers in general, given the low wages and volatility associated with nonstandard work schedules. Previous research has documented cross-national differences in the work-family conflict of nonstandard work schedules among partnered workers in Europe (e.g., [53]). However, because parents generally experience higher levels of stress and financial strain than childless individuals, these differences may amplify the negative health consequences among parents compared to non-parents. In addition, such links might differ for parents due to institutional provisions that are particularly relevant to them, such as childcare policies. Studies have shown that childcare costs are an important reason parents choose nonstandard work schedules [32,33,52]. If so, we might expect that the links between parental work schedules and health differ across countries, given that institutional provisions may shape parental choices about work schedules by making childcare available and/or affordable, and, in turn, benefiting their well-being. Indeed, studies have shown that the links between insecure and unstable jobs and workers' health depend partly on labor market institutions (e.g., working time regulations) and social protection provisions (e.g., income support policies) [20]. By extension, we expect that differences in availability and affordability across countries may shape the links between parental work schedules and parental health.

The arguments and findings discussed above suggest that the association between nonstandard work schedules and health may differ between parents and the general population. On the one hand, parents might experience stronger negative health effects than childless individuals due to additional family responsibilities and the greater challenges of balancing work and family life. On the other hand, nonstandard work schedules may offer advantages for some parents by allowing more flexibility in coordinating childcare and household duties, potentially easing work–family conflict. Yet again,

the national differences in the social, economic, and policy contexts, such as the availability and affordability of childcare, may further complicate the links between work schedules and parental health.

Thus far, most existing studies are based on U.S. samples and produced mixed findings. For example, a study based on a 1992 national workforce sample from the US found parental distress, burnout, and work-family conflict were strongly associated with lacking scheduling control but not with nonstandard shifts themselves [54]. Using U.S. data, Perry-Jenkins et al. [3] found a strong association between evening or night work schedules and increased levels of depression among new mothers and fathers, whereas Grzywacz et al [55] as well as Shepherd-Banigan et al. [56] found no significant relationship between maternal work schedules and depression during the first two years of a child's life, although Shepherd-Banigan et al study found a significant association between working from home and lower depression scores. Using a nationally representative dataset of parents from Australia, Zhao et al. [19] found no association between nonstandard work schedules and parental psychological distress. In contrast, using an online community-based sample, Zhao et al. [57] found a significant association between irregular shifts and parental psychological distress. Likewise, a recent study using a longitudinal birth cohort in the UK found a strong association between fathers' evening and weekend schedules and poorer mental health for fathers [7]. These mixed findings might be due to various reasons. One reason might be that, regardless of the motivation for the work arrangements, for any parent, having nonstandard work hours may introduce stressors into life due to misaligned family routines and schedules [8,58], placing an unavoidable physical toll on workers and their biological functioning [9]. The second might have to do with the nature of the data (e.g., cross-sectional vs. longitudinal), sample representation (e.g., local vs. nationally representative), and the aspects of nonstandard work schedules (e.g., volatility measured by lacking scheduling control vs. a nonstandard shift when carried out at home). The findings thus far paint a mixed picture regarding the nature of the data: both longitudinal [3 vs. 55] and cross-sectional [57 vs. 54] data have shown both significant and non-significant associations, and whether these data were nationally representative or a locally non-representative sample. The findings seem more consistent when it comes to specific aspects of nonstandard work schedules (e.g., lack of scheduling control [57] vs. ability to work from home [56]). Another reason for the mixed findings might be due to samples from different contexts, where societies with varying employment and family policies may not only shape the prevalence of nonstandard work schedules but also alter the associations between work schedules and parental health [7], making findings from one country challenging to apply to other countries. Our cross-country comparative analysis using nationally representative and longitudinal data that delineate dynamic changes in work patterns spanning three decades may thus shed informative light on the links between parents' longitudinal work schedules and later health across countries with varying labor market and institutional provisions.

## The present study

This study investigates whether parental work schedules are associated with later health and well-being, and whether this association varies across different national contexts. We draw upon four longitudinal datasets from four developed countries—Australia, Germany, the UK, and the US —to systematically examine how work trajectories during prime working ages are associated with physical and mental health outcomes at various stages of adulthood (ages 35/40, 45/50, and 55/60). Our study contributes to the state of knowledge by (1) adopting a life course perspective to carefully chart parental work trajectories between their 20s to 50s, (2) adopting a cross-country lens to understand if the association between parental work schedules and health might be context driven, and (3) utilizing the sequence analysis to not only properly chart the dynamic changes in work arrangements but also create proper temporal sequence between exposure and outcome, adding a critical empirical evidence of parental work schedules and parental health for which we only have a handful of empirical evidence thus far. We have chosen these four countries because national longitudinal data on parental health, along with their work schedules, are readily available and comparable between these datasets, which is rare in comparative studies. We thus acknowledge that while the empirical findings from this study are informative and may be generalizable to similar economies and societies, they may not apply to countries with drastically different

policy environments or economies. This study serves as a vital step in advancing our understanding of intergenerational health equity and disparities in parental health, as parental health carries important implications for child development and well-being, ultimately affecting our future workforce [59].

## Materials and methods

### Data

S1 File presents the details of each country's data used for the analysis. Specifically, we use (1) the Household, Income and Labour Dynamics in Australia (HILDA) Survey for Australia, a household-based panel study that collects longitudinal information from about 17,000 respondents aged 15 and over annually on economic and personal well-being, labour market dynamics, and family life since 2001; (2) the German Socio-Economic Panel (SOEP), a household-based panel study initiated in 1984, which interviews around 30,000 individuals each year on a wide range of socio-economic topics; (3) the Household Longitudinal Study (UKHLS) for the UK, started with 40,000 households in 2009–10 with now 12-wave data thus far; and (4) the National Longitudinal Survey of Youth-1979 (NLSY79) for the US, with an initial sample of 12,000 individuals ages 14–22 in 1979 to collect rich sociodemographic, labor market, and health and well-being information annually and then biennially since 1994. These secondary data are publicly available and exempt from IRB ethics review.

### Data comparability

To ensure data compatibility and to accommodate the slight variations in data collection methods by country, we make several choices. First, we chart parental work schedules by focusing on three age periods that are consistent across all four countries: 25–34, 35–44, 45–54; these three time periods proxy the family life stages (e.g., starting career, marriage, and parenthood). Hence, we include participants who indicated they became or were parents during the corresponding 10-year period. NLSY79 is a longitudinal cohort study of American youth aged 14–22 in 1979 and thus identifying parents over the years is straightforward. However, because the data collection structures administered by HILDA, SOEP, and UKHLS also sampled individuals in mid- and late-life, we retain all participants who had 7 or more consecutive years of data, allowing us to chart their work trajectories more precisely within each of the 10-year time periods. Thus, we note that the presence of "missing" in the cluster solutions for these three countries may largely result from the data structure that the majority of parents were observed for about 7–8 years instead of 10 straight years, leaving us with no data for the last two years or so (hence, a "missing" state). Second, we examine the health outcomes available across countries at three time points: 35/40, 45/50, and 55/60, allowing us to investigate, for example, the relationship between parental work schedules during ages 25–34 and health outcomes at age 35. We note that whereas the health outcomes were available at ages 35, 45, and 55 for Australia, Germany, and the UK data, the health outcomes were collected through health modules at ages 40, 50, and 60 for the US data; we are cautious about the age differences but also value the comparability of these four datasets. Fourth, we use age 25 as the starting point to document work schedule patterns over one's working years for participants across countries would have begun their careers by that point. Fifth, we use a common classification of work schedule variables to ensure consistency between countries: not working, working standard daytime hours, evenings (3–9 pm), nights (9 pm-6 am), or irregular hours or weekends. We combine irregular hours and weekends as the US data did not include questions about weekends, and the data from Germany and the UK included questions about weekends but not irregular hours. Sixth, we consider a wide range of sociodemographic characteristics available across countries.

### Measures

S1 Table presents the details on all analyzed variables, including how we measure and define each variable. We examine a range of health outcomes including self-assessed general health status (poor/fair vs. otherwise), physical health

(SF-36/ SF-12 physical function), mental function (SF-36/ SF-12 mental function), and depression/psychological distress (Kessler-10 for the Australian samples; GHQ-12 for the UK samples; CES-D for the US samples). Previous studies have demonstrated that the three scales of Kessler-10, GHQ-12, and CES-D are comparable [60,61]. For easy comparison and interpretation of effect sizes, we standardize SF12/SF36 to have a mean of 50 and a standard deviation of 10, so that a one-point difference in the regression estimates corresponds to a one-tenth of a standard deviation effect size, which is comparable across countries. We also include a comparable set of sociodemographic characteristics measured before the outcome ages (i.e., 35/40, 45/50, 55/60): gender (female vs. male), age, race-ethnicity or migration background (where the concept applies to the country context), education, partnership status, and parenthood status, and employment characteristics between ages 25–34, 35–44, and 45–54 (primary occupation and average weekly working hours). To define primary occupation during the 10-year period, we calculate the share of the number of years the participant has engaged in that particular occupation, and the primary occupation is defined when that specific occupation has occurred at least half of the 10-year period; when the share of the number of years is approximately equal, we define the participant to have a mixed occupation. We use the same calculation method to define the primary weekly working hours as either primarily full-time (>=35 hours per week), primarily part-time (1–34 hours per week), or an equal share of full- and part-time hours.

## Data analysis

Missing rates are about 5% or lower for sociodemographic variables and our work schedule variables. Following the scholarship, we use a complete case analysis to address missing data for the sociodemographic and work schedule variables due to their low missing rates, i.e., <=5% [62]. Higher missing rates are observed for the outcome variables, and we do not conduct multiple imputation, as recommended by previous scholarship, to avoid measurement error [63]. S3 Table provides details on the descriptive statistics for each analyzed variable, along with the number of missing cases for each age period and country. As a result of excluding cases with missing data on sociodemographic variables and outcome variables, the final analyzed sample sizes for each dataset and outcome at the three outcome age periods are approximately 1,500 for HILDA, SOEP, and UKHLS, and around 5,000 for NLSY79.

We conduct our analysis in two steps. First, we use sequence analysis to chart the work trajectories for each country by the age groups of 25–34, 35–44, and 45–54. Sequence analysis is ideal for examining work trajectories through a life course lens, as it captures transitions in categorical variables over time [64]. The categorical variable in this case is the work schedule variable with five "states" (i.e., not working [NW], standard daytime [ST], evenings, nights, and other nonstandard schedules [other NST]) along with the time parameter (i.e., ages 25–34, 35–44, 45–54) for classification. We also include "missing" as the sixth state to account for occasions when respondents did not answer the question despite being interviewed or when respondents were not part of that survey year. Each participant's state of work schedule progression over this time frame is called a "sequence." Utilizing the optimal matching algorithm to assess the dissimilarities and similarities by setting the "costs" associated with transforming one sequence into another [65,66], the sequence analysis groups these sequences, or the static or changing states over time, into clusters. In line with prior research using the Needleman-Wunsch algorithm [67,68], we set the insertion and deletion costs to 1 to calculate the substitution costs based on the transition rates between work schedule states. Subsequently, we employ Ward's hierarchical fusion algorithm to group similar sequences into a more concise set of clusters [65]. We apply stopping rules based on the Duda-Hart index, the Calinski-Harabasz pseudo-F index, and the conceptual relevance of the clusters [69] to determine the optimal number of clusters. In general, the larger the values of these fit statistics, the more ideal the cluster solutions are. We provide these diagnostic test results in S2 Table. Our sequence analysis answers the following two research questions: (1) How do parental work patterns evolve over the life course? And (2) Are there cross-country differences in parental work patterns?

Next, based on whether the outcome variables are continuous or dichotomous, we utilize ordinary least squares (OLS) or logistic regression models to investigate the extent to which work schedule patterns between ages 25–34, 35–44, and

45–54 are associated with individual health outcomes at ages 35/40, 45/50, and 55/60, respectively. This part of the analysis answers the following two research questions: (3) Are certain work patterns associated with worse health outcomes than others? And (4) Does the strength of these associations vary across countries? In each analytic model, we control for the rich set of sociodemographic characteristics detailed in the Measures section and in S1 Table. We use Stata v.18 for all analyses.

## Results

### How do parental work patterns evolve over the life course and differ by country?

S2 Table presents the details of the three diagnostic indices discussed in the Data Analysis section. S1-S3 Figs display the cumulative distribution of the cluster solutions obtained from the sequence analysis. For brevity, Table 1 summarizes the work patterns by age and country. In general, parents across countries have engaged in various work patterns during their prime working years, shifting from four dominant trajectories during the 25–34 age range to more diverse profiles starting in the mid-30s. The common trajectories include (1) mostly not working (NW) or transition into a combination of NW and standard-nonstandard (ST-NST) (i.e., ST plus NST schedules), (2) mostly other NST (e.g., weekends, irregular hours), (3) mostly Evenings or Nights, (4) a combination of ST + NST, and (5) mostly ST or ST only. Generally, about half of the parents tend to work stable standard daytime hours (ST) throughout these three 10-year periods, from ages 25–54, with another half of the parents engaged in a decent degree of nonstandard work schedules and volatile patterns (e.g., frequent switching between different work statuses and schedules). The prevalence of parents working nonstandard

**Table 1. Summary of parental work patterns by age and country.**

| | Australia | Germany | UK | US |
|---|---|---|---|---|
| Between Ages 25–34 | 9%: Mainly NW | 22%: Mainly NW to some ST+NST | 26%: Mainly NW to some ST+NST | 16%: Volatile to mainly NW |
| | 20%: NW+ST | 18%: Nights+other NST | 20%: Volatile to mainly other NST to volatile | 13%: Volatile to mainly other NST |
| | 21%: Other NST+ST | 16%: Volatile to mainly evenings | **31%: Mainly ST+some NST** | 25%: Volatile |
| | **50%: Mainly ST** | **44%: Mainly ST** | 23%: Mainly ST | **45%: Mainly ST** |
| Between Ages 35–44 | 13%: Mainly NW | 8%: Mainly NW | 21%: Mainly NW to some ST+NST | 13%: Mainly NW to some ST |
| | 22%: Volatile to mainly ST | 24%: Evenings+other NST | 11%: Mainly Evenings to volatile | **43%: ST+NST** |
| | 7%: Mainly other NST | 11%: Mainly nights | 17%: Mainly other NST to ST+NST | 9%: Mainly other NST |
| | **33%: Mainly ST** | **30%: Volatile to mainly ST** | 22%: Mainly ST to missing | 35: ST only |
| | 24%: ST only | | **28%: Mainly ST** | |
| Between Ages 45–54 | 10%: Mainly NW | 16%: Mainly NW to some ST | 12%: Mainly NW | 8%: Mainly NW |
| | 8%: Volatile | 13%: Mainly nights | 31%: Mainly ST+some NST | 18%: ST+NST to mainly ST |
| | 10%: ST+other NST | 10%: Mainly evenings | 18%: Mainly other NST+evenings | 7%: ST to other NST |
| | 9%: Mainly other NST | 14%: Mainly other NST | **40%: Mainly ST** | 6%: Other NST to ST |
| | **28%: Mainly ST** | 13%: Mainly ST to ST+NST | | 11%: Mainly evenings/nights |
| | 8%: Mainly ST to missing | **34%: Mainly ST** | | 10%: Mainly other NST |
| | **27%: ST only** | | | **39%: ST only** |

*Note.* NW: not working; ST: standard daytime hours; NST: nonstandard hours; other NST: weekends/irregular hours. Bolded categories represent the largest group, while the underlined represent patterns involving a certain degree of NST along with volatility.

schedules (e.g., irregular hours, weekends) is higher during the childbearing and childrearing years (e.g., 35–44, 45–54) than in other life course stages.

Specifically, a considerable share of parents in their 20s start with a not-working (NW) status (ranging from 16% in the US to 22% and 26% in Germany and the UK and 29% in Australia). About half of the parents work in mostly stable standard daytime hours in every country. Worthnoting is a substantial share of parents engage in nonstandard work schedules or volatile patterns (frequent changes between different work schedules), ranging from 20% and 21% in the UK and Australia to 34% and 38% in Germany and the US.

Whereas the majority (≥50%) of the parents are still engaged in stable standard daytime schedules during their mid-30s to early 40s, we also observe a greater share of parents show more diverse work profiles involving in a combination of working ST, evenings, nights, and other NST, ranging from 29% in Australia, to 35% in Germany, 49% in the UK, and 52% in the US. For example, on the one hand, in the UK, we observe that 21% of parents had a work pattern that changes from primarily not working to a combination of standard and nonstandard hours (i.e., evenings or nights), classified as "mainly NW to some ST + NST." Another 11% of parents have a work pattern of shifting from mainly evening hours to a volatile schedule (e.g., changing between NST and ST, etc.), classified as "mainly Evenings to volatile." Another 17% of parents have a work pattern of switching from mainly other NST (weekends or irregular hours) to a combination of ST and NST (i.e., evenings and nights), classified as "mainly other NST to ST+NST." On the other hand, in the US, 43% of parents have a work pattern combining ST with NST, and another 9% mainly have irregular hours (other NST).

Between their mid-40s and early 50s, we observe an equal, if not more diverse, range of work patterns among parents in these four countries. About 63% of Australian parents in their mid-40s to early 50s have work patterns of either "mainly ST" or "ST only," with another 27% engaging in some degree of volatile or nonstandard work schedules. In contrast, about 40% of UK parents have a "mainly ST" work pattern, but nearly half of UK parents (31% + 18%) have work patterns of combining ST, NST, and other NST. Similarly, about 39% of US parents have a "ST only" work pattern with another roughly 30% of parents working a variety of nonstandard work schedules (7% + 11% + 10%). Notably, about a quarter of US parents switch their nonstandard work schedules to standard daytime hours (18% + 6%). Lastly, about 34% of German parents have a "mainly ST" work pattern, but half of the parents (13% + 10% + 14% + 13%) work a variety of nonstandard work schedules or a combination of ST and NST.

Despite these similarities in work patterns over the three age periods across countries, we note differences that might have to do with the working time flexibility and regulations in each country. For example, during their 20s and early 30s, about a quarter of respondents in Germany and the UK, where working time flexibility is guaranteed by law and where there is more generous parental and child support, switch their work patterns from not working to a combination of ST and NST, whereas 16% of US respondents switch from volatile to not working. This subtle difference might suggest the challenges facing US respondents juggling work and family demands, who opt out of the labor force, perhaps because high childcare costs outweigh the income a job could bring in. Such challenges are exacerbated by the lack of working time flexibility and adequate family support in the US. During their mid-30s to early 40s, we continue to observe about 20% of UK parents shifting from not working to a combination of ST and NST while only 13% of US parents do so. We also observe that 30% of German parents switch from volatile work patterns to stable standard daytime hours in their mid-30s to early 40s, which we believe is challenging for US parents to achieve, as volatility in the US labor market tends to be associated with workers from relatively disadvantaged backgrounds in a country with weak labor market regulation and social protection provisions.

## Sociodemographic profiles by work patterns

S3 Table presents the descriptive statistics for all analyzed variables for the total sample, as well as by sequence cluster solution of work patterns, by age and country. In general, these descriptive statistics paint a consistent picture of the

sociodemographic characteristics of parents who are more or less likely to have stable standard daytime hours, as well as those who are more or less likely to have nonstandard work schedules or schedules involving volatility and uncertainty. White (or non-immigrant or non-indigenous) males with high educational levels and professional/managerial occupations are more likely to have stable standard daytime hours (i.e., "mainly ST" or "ST only"). In contrast, females, those of Blacks or with migration or indigenous backgrounds, of low educational levels, working primarily non-professional/non-managerial occupations, are more likely to be not working or working at nonstanadard or volatile schedules. Parents with medium educational levels or of Blacks (in the UK and US settings) are also more likely to have work patterns characterized by nonstandard work schedules with volatility. Not surprisingly, parents who are mainly not working have the poorest health outcomes, while those with stable daytime hours have the best health outcomes.

A few differences across these four countries are worth mentioning. First, about half of the participants in the Australian and US data are females. In contrast, about two-thirds of the participants in Germany and the UK are females. Hence, we observe that when participants identify themselves as parents during those 10-year periods in these two countries, the overwhelming majority (ranging from 75% in the 25–34 age range to more than 95% in the 35–44 age range) report not working. Second, parents in Germany in a professional or managerial capacity are more likely to work evenings or nights with full-time hours, or the longest average working hours among all groups. We also observe that these parents tend to have the best health outcomes among all groups. Third, parents in Germany and the UK transition from full-time to mainly part-time hours once they reach the ages of 45–54. As a consequence, in many ways, parents in Australia and the US are relatively similar compared to Germany and the UK in terms of the descriptive associations between work patterns and their sociodemographic characteristics: Australian and US parents work much longer hours (throughout), with the ST profiles being the staple for those with relatively advantaged sociodemographic characteristics.

### Are certain work patterns associated with worse health outcomes than others?

S4 Table presents the details of the multiple regression estimates of parental work patterns on health outcomes by age and country. For brevity, Table 2 summarizes the statistically significant regression estimates. In general, we find statistically significant associations between various parental work patterns and health outcomes across the three age stages. However, the significant (adverse) associations seem to be more prevalent and stronger at later ages, especially at ages 55/60. Specifically, parents with a "mainly NW" work pattern or work patterns involving a fair share of NW report significantly poorer health outcomes at all three age stages for all countries, ranging from two-tenth of a standard deviation lower on physical and mental functions for parents in the UK and the US at ages 35/40 to eight-tenth of a standard deviation and a half standard deviation lower on physical and mental functions, respectively, for parents in the UK at age 55. Furthermore, engaging in some degree of nonstandard or volatile schedules is associated with significantly poorer health for parents across all four countries, and this is particularly true for health outcomes at ages 55/60. The effect sizes range from one-tenth of a standard deviation in the US to three-tenths of a standard deviation in the UK ("mainly NW to some ST+NST" at age 45 on physical function) and Germany ("mainly nights" at age 55 on physical function) and to a half standard deviation in Australia (i.e., "volatile" work pattern at age 55 on physical function). Across all four countries, parents who have either "mainly NW" or some degree of volatile or nonstandard work schedules are more likely to report having poor/fair health, higher scores for psychological distress/depression, and a higher likelihood of having at-risk psychological distress/depression than those with a "mainly ST" work pattern.

We also conduct interaction analysis to test if the reported associations might be more pronounced for either mothers or fathers. The existing research suggests that mothers are more likely to shoulder the physical and mental health burdens in the context of work arrangements than fathers [56,70]. Our interaction analyses reveal several statistically significant differences between fathers and mothers. Specifically, fathers are more likely than mothers to fare poorly in

Table 2. Summary of multiple regression estimates of parental longitudinal work patterns on health outcomes.

| Ref: Mainly ST | SF36 or SF12 Physical Function | SF36 or SF12 Mental Function | Self-Assessed Poor/Fair Health | Psychological Distress/ Depression score | At risk of psychological distress/ depression |
|---|---|---|---|---|---|
| **Australia** | | | | | |
| Age 35: Mainly NW | −4 | −4 | + | + | + |
| Age 35: NW+ST | −2 | −3 | + | + | + |
| Age 45: Mainly NW | −4 | −2 | + | + | + |
| Age 55: Mainly NW | −4 | −4 | | | |
| Age 55: Volatile | −5 | −3 | + | + | |
| Age 55: ST only | +2 | +1 | − | − | |
| **Germany** | | | | NA | NA |
| Age 35: Mainly NW to some ST+NST | | | + | | |
| Age 45: Mainly NW | | | + | | |
| Age 55: Mainly NW to some ST | −5 | | + | | |
| Age 55: Mainly Nights | −3 | | + | | |
| Age 55: Mainly Evenings | −2 | | | | |
| **UK** | | | | | |
| Age 35: Mainly NW to some ST+NST | −2 | | | | |
| Age 35: Mainly ST+some NST | −2 | | | | |
| Age 45: Mainly NW to some ST+NST | −3 | −2 | + | + | |
| Age 55: Mainly NW | −8 | −5 | + | + | + |
| **US** | | | | | |
| At age 40 | | | | | |
| Volatile to mainly NW | −2 | −2 | + | + | + |
| Volatile to mainly other NST | −1 | | + | + | |
| Volatile | −1 | −1 | + | + | + |
| At age 50 | | | | | |
| Mainly NW to some ST | −2 | −2 | + | + | + |
| ST+NST | −1 | −1 | + | + | + |
| At age 60 | | | | | |
| Mainly NW | −4 | | | | |
| ST+NST to mainly ST | −1 | | | | |
| ST to other NST | −2 | | | + | + |
| Mainly Evenings/Nights | | −1 | + | + | |
| Mainly other NST | −3 | | | + | + |

*Note*. Ref.: Reference group; NA: Not available. Only the cells where a statistically significant association found were displayed with numbers or symbols. The numbers represent the multiple regression estimates shown in S4 Table; for example, "-4" represents a four-tenth standard deviation difference in the corresponding health outcomes when parents had that particular work pattern compared to the reference group, which had a "mainly ST" work pattern. The symbol of "+" represents a significant positive association, whereas "-"represents a significant negative association. Different color highlights in the work patterns were intended to distinguish the health outcomes at different age periods.

physical and mental functions when they have either "mainly NW" or work arrangements with a fair share of NW (e.g., "mainly NW to some ST+NST" or "mainly NW to some ST"). However, mothers are more likely to have poor physical or mental health than fathers when they engage in work arrangements characterized by volatility (e.g., "volatile to some ST", "mainly other NST", or "volatile to other NST"). Nonetheless, we are mindful that these statistically significant interaction effects may be due to chance, given the large number of interaction terms analyzed (20 of 211 are significant).

## Does the strength of these associations vary across countries?

We observe consistently poorer health outcomes among parents who have either "mainly NW" or some degree of volatile or nonstandard work schedules than their counterparts across countries. Our results also show that the adverse health consequences seem to be more persistent from early to late ages (ages 40–60) in the US than in the other three countries, but the adverse health consequences have larger effect sizes (i.e., physical and mental functions) in Australia, Germany, and the UK than in those observed in the US. The consistent, persistent adverse association across countries between not working (or in combination with nonstandard work schedules) and later poor health outcomes among parents may suggest that not working is associated with relative disadvantages that limit parents' access to health-promotion resources, a pattern uniformly experienced by parents [71]. Worthnoting is that the significant adverse physical functions reported by German parents at age 55 are not accompanied by similar adverse mental function outcomes, in contrast to the other three countries, possibly suggesting, on the one hand, German parents working nonstandard schedules or a mixture of unstable schedules including not working may not feel stressed and distressed as parents in other countries due to more adequate family support and work flexibility. On the other hand, German parents are just as exhausted and fatigued as their counterparts in Australia, the UK, and the US, with work patterns that are anything but stable standard daytime hours, suggesting such work arrangements may weigh heavily on our physical functions when we age. Importantly, we contemplate if the generous support for labor market and social protection provisions may shield German parents from potential psychological harms due to work schedules. Our conjecture is reinforced by the finding of persistent adverse health outcomes among parents with relatively volatile work trajectories in the US (and to a lesser degree in Australia), suggesting that unstable and unpredictable work patterns might be particularly detrimental to health in societies with weak labor market regulation and social protection provisions.

## Discussion and conclusion

Using four longitudinal datasets from four developed Western economies (Australia, Germany, the UK, and the US), we investigate the links between parental work trajectories throughout three pivot age periods and their later health outcomes. Our empirical findings suggest five important findings. First, parents across these four countries engage in a variety of work arrangements during their prime working years, and these arrangements seem to involve more combinations of various work schedules (e.g., ST+NST, ST+other NST, etc.) during the mid-30s to mid-40s than during the mid-20s. Second, parents who engage in work patterns of mainly not working or in combination with other work schedules throughout their prime working ages (i.e., 25–54) fare the worst in terms of their physical and mental health, and this is true across all four countries. Third, parents with any work schedules involving a certain degree of volatility (e.g., changing between schedules) experience significantly worse health than those with stable daytime hours. Fourth, the above observed adverse health consequences are persistent from early to later ages, with effect sizes ranging between two-tenth standard deviation to half a standard deviation for physical and mental functions. Fifth, the effect sizes of these adverse health consequences are comparably larger for parents in Germany and the UK at later ages, but are more persistent for parents in Australia and the US. Indeed, we observe that parents in Australia and the US experience the most consistent negative effects across a wide range of health outcomes. Our empirical results strongly suggest that parental work arrangements

may carry long-lasting impacts on parental health. Below, we discuss a few critical results corresponding to these five points.

First, we observe that parents seem to engage in more diverse work patterns that might correspond to the life course transition into marriage and childbearing and rearing during the mid-30s to early 40s. Although our study does not directly investigate the possibility of whether these work arrangements coincide with family arrangements due to marriage and childbearing, a recent empirical study using the US data has explicitly examined the links between work and family arrangements and has found that couples with different numbers of children (e.g., two vs. three or more) and at different life stage (e.g., early vs. later ages) engage in significantly different work patterns [72]. For example, couples having two children in their late 30s are more likely to have stable standard daytime schedules compared to those having three or more children earlier in their life (late 20s to early 30s) who are more likely to have volatile work schedules (e.g., switching between not working to working, between nonstandard hours to not working). Our empirical findings suggest that parents in these countries are likely to juggle demands from both work and family, as parents across these four countries engage in similar work arrangements during their 30s and early 40s.

We also note that parents in Australia and the US experience the most consistent negative effects of work patterns characterized by either not working or volatile arrangements across a wide range of health outcomes and over time, yet the magnitudes of these negative effects are stronger (especially at age 55) for parents in Germany and the UK. One would predict that individuals in a society with weaker and fewer social protection measures (e.g., decent wages, healthcare benefits) are likely to suffer more health-wise compared to those in societies with stronger and better social protection measures [73]. If so, we would expect that parents in the US, which provides weaker or no social protection for their workers, would experience the most adverse health consequences when engaging in work patterns that are anything but stable and standard daytime hours. Our results, however, indicate that parents in Germany and the UK experienced a larger magnitude of adverse health consequences in later adulthood (at age 55) when engaging in work patterns other than stable, standard daytime hours. Although it is beyond the scope of this paper, we draw upon and extend the SDOH argument on how social determinants (e.g., living conditions, working conditions, quality of food and housing, etc.) might influence health [74]. Specifically, the institutional lens discussed in our literature review may help explain the persistent adverse associations observed in Australia and the US, due to weaker family and social protections, higher exposure to volatile schedules, and fewer mechanisms for workers to negotiate predictable hours. In contrast, the larger magnitude of associations observed in Germany and the UK might align with psychosocial comparison mechanisms posited in the broader social determinants of health. In his elaboration, Bartley theorized that psychosocial comparison is an essential component of how social determinants influence health. Specifically, the psychosocial comparison view posits that, amidst social determinants (e.g., living and working conditions), people may compare themselves to others, and these comparisons can influence how individuals perceive their health and well-being. If so, the weaker effect sizes observed for US parents might have to do with the overall comparison that everyone is worse off in a society with weaker or no social protection. As a result, parents may not feel drastically different from others, but still experience significantly worse health outcomes. Similarly, the large effect sizes observed for German and UK parents might be due to the comparison that, on average, people are doing well, likely due to strong social protection measures. Consequently, the accumulation of having work patterns that are anything but stable standard daytime hours might weigh heavily on their self-reported long-term health compared to that of others; this is demonstrated by a large effect size observed at age 55. These institutional contrasts underscore that the long-lasting health effects of parental work arrangements are shaped not only by individual characteristics but also by the regulatory, economic, and social protection systems in which those trajectories unfold.

## Limitations

Despite our extensive effort, our observational study is bound to have limitations. First, we have relatively small sample sizes for Germany at age 35, which might hamper the identification of significant findings among this age group. This limited sample size is in part due to the relatively late labor market entry of college-educated individuals [75], so not all

respondents have complete ten-year employment histories available. However, the sample sizes for the age groups of 45 and 55 are comparable to those in Australia and the UK, although they are smaller than those in the US, allowing meaningful cross-country comparisons. Second, although we are careful to measure variables consistently across all four countries' data, fully addressing the data comparability issue remains challenging. For example, US data do not distinguish weekends from weekday nonstandard schedules, whereas the other three countries do. We thus combine weekend work with irregular hours as "other NST." However, this categorization risks masking the effects of irregular hours from those of weekend hours and would be particularly problematic if one schedule is beneficial to health while the other is not. Hence, if anything, our measure of "other NST" might be underestimated. Furthermore, we conduct separate regression analyses for each country rather than pooling the datasets, allowing us to examine the relationship between work patterns and health outcomes within each country rather than across countries, thereby avoiding potential measurement noise from cross-country data harmonization. However, we acknowledge that this approach has its own limitations; for one, it cannot use coefficients from the same model to properly compare associations across countries. Hence, we present our findings with effect sizes to address comparability issues arising from coefficient differences. Third, and relatedly, despite our efforts to make the measurements of work schedules and health outcomes as comparable as possible, cautions are needed when interpreting the effect sizes between countries, partly driven by the possibility that people in different societies may hold various understandings relating to various work schedules and the questions regarding personal physical and mental health. Fourth, due to the data at hand, we could not distinguish whether individuals (in)voluntarily chose specific work schedules, as the scholarship suggests that voluntarily choosing specific work schedules might produce benefits that outweigh the physical and psychological tolls [32,58]. If so, our results are likely to be biased downwards, as our analyses do not account for the beneficial effects of voluntarily choosing specific work schedules on health outcomes. Fifth, we acknowledge that parents may switch work statuses or schedules due to worsened health from work or working nonstandard hours. If so, the potential negative health effects of nonstandard or volatile work schedules might be partly absorbed into the negative effects of "not working", creating the possibility that our effect sizes might be anything but underestimated. Sixth, we utilize self-reported health outcomes that are standard and have been widely used in various disciplines (e.g., public health, psychology, sociology), facilitating the comparability between our empirical findings and the state of knowledge on these health outcomes. On the one hand, assessing our own health provides a valuable subjective perception of how we are doing that no other measure can compete with. On the other hand, we need to acknowledge that this subjective perception may be influenced by recall or social desirability biases. Last but not least, establishing causal links between parental work schedules and parental health is impossible given the non-experimental nature of the topic. However, the extant research has identified why having nonstandard or volatile work patterns might be harmful to health due to its poor employment conditions such as irregular schedules, insufficient pay, and limited or no benefits [2,9], staple features of lacking access to health-promoting resources. Hence, policies to address the health harming aspects of work patterns are promising avenues to tackle the causality issue. For example, establishing an experimental design to raise low wages to living wages or to provide essential benefits (e.g., health insurance, pensions, paid sick leave) holds promise to provide evidence of the causal health effects of volatile or nonstandard work patterns [76]. We also have a decent body of scholarship on how implementing family-friendly policies such as flexible work arrangements (e.g., working from home) may enhance workers' and parents' health [77,78].

## Conclusion

Despite these limitations, our study marks a critical step in producing a cross-country comparison of the links between parental long-term work trajectories and their later health outcomes, and demonstrates the diverse work arrangements juggled by parents across these four Western countries. The persistent negative associations found for Australian and US parents throughout the early to late ages and the large negative associations found for German and UK parents at later ages on their health outcomes speak volumes to how work might have become a vulnerability to parental health, particularly when the arrangements were anything but stable and standard daytime hours. We do not foresee in the near

future that our labor market domestically and globally will be reversed to produce more decent and stable employment for everyone. As a large body of scholarship has demonstrated the close causal links between parental health and well-being and their children's health and well-being [18], our results carry critical implications for how parental work trajectories might become a vital vehicle for perpetuating intergenerational social and health disparities. Solid evidence shows that employment characterized by stable hours, adequate compensation, and job-related benefits is crucial for achieving and maintaining good health and well-being [79], with implications for future generations' well-being. Indeed, studies examining US policy changes over the past 25 years have provided strong evidence that income support policies (e.g., expanded Earned Income Tax Credit), when paired with good jobs, may enable families, especially those facing disadvantages, to achieve consistent positive child well-being, thereby reducing intergenerational transmission of poverty and inequality [18].

## Supporting information

**S1 Table. Measures and variable comparability.**
(DOCX)

**S2 Table. Work arrangement sequence cluster solution diagnostic indices.**
(DOCX)

**S3 Table. Descriptive statistics of analyzed variables by country and age.**
(DOCX)

**S4 Table. Parental health outcomes by age and country.**
(DOCX)

**S1 Fig. Sequence analysis distribution plot for work arrangement among parents aged 25–34.**
(DOCX)

**S2 Fig. Sequence analysis distribution plot for work arrangement among parents aged 35–44.**
(DOCX)

**S3 Fig. Sequence analysis distribution plot for work arrangement among parents aged 45–54.**
(DOCX)

**S1 File. Data details.**
(DOCX)

## Author contributions

**Conceptualization:** Wen-Jui Han, Matthias Pollmann-Schult, Tinh Doan, Jianghong Li.

**Data curation:** Wen-Jui Han, Johanna Carrasco Saravia, Matthias Pollmann-Schult, Tinh Doan, Jianghong Li.

**Formal analysis:** Wen-Jui Han, Johanna Carrasco Saravia, Matthias Pollmann-Schult, Tinh Doan.

**Investigation:** Wen-Jui Han, Matthias Pollmann-Schult.

**Methodology:** Wen-Jui Han, Matthias Pollmann-Schult, Tinh Doan,  Jianghong Li.

**Project administration:** Wen-Jui Han.

**Software:** Wen-Jui Han, Johanna Carrasco Saravia.

**Supervision:** Wen-Jui Han, Matthias Pollmann-Schult.

**Validation:** Wen-Jui Han.

**Visualization:** Wen-Jui Han, Johanna Carrasco Saravia.

**Writing – original draft:** Wen-Jui Han.

**Writing – review & editing:** Wen-Jui Han, Matthias Pollmann-Schult, Tinh Doan, Jianghong Li.

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
