## [Decision Letter · Decision Letter 0]

26 Jan 2026

PONE-D-25-46936Longitudinal Employment Patterns and Parental Health: A Cross-Country LookPLOS One

Dear Dr. Han,

Thank you for submitting your manuscript to PLOS ONE. After careful consideration, we feel that it has merit but does not fully meet PLOS ONE’s publication criteria as it currently stands. Therefore, we invite you to submit a revised version of the manuscript that addresses the points raised during the review process.

We look forward to receiving your revised manuscript.

Kind regards,

José Alberto Molina

Academic Editor

PLOS One

Journal Requirements:

3. Please include a copy of Table 1 and 2 which you refer to in your text on page 16 and 20.

5. We note that there is identifying data in the Supporting Information file “S1 Table_PLOS supporting information.docx, S3 Table_PLOS supporting information.docx and S4 Table_PLOS supporting information.docx”. Due to the inclusion of these potentially identifying data, we have removed this file from your file inventory. Prior to sharing human research participant data, authors should consult with an ethics committee to ensure data are shared in accordance with participant consent and all applicable local laws

-Location data

6. If the reviewer comments include a recommendation to cite specific previously published works, please review and evaluate these publications to determine whether they are relevant and should be cited. There is no requirement to cite these works unless the editor has indicated otherwise.;

Reviewer's Responses to Questions

**Comments to the Author**

1. Is the manuscript technically sound, and do the data support the conclusions?

Reviewer #1: Yes

Reviewer #2: Yes

2. Has the statistical analysis been performed appropriately and rigorously? 

Reviewer #1: Yes

Reviewer #2: Yes

3. Have the authors made all data underlying the findings in their manuscript fully available?

Reviewer #1: Yes

Reviewer #2: Yes

4. Is the manuscript presented in an intelligible fashion and written in standard English?

Reviewer #1: Yes

Reviewer #2: Yes

5. Review Comments to the Author

Reviewer #1: Longitudinal Employment Patterns and Parental Health: A Cross-Country Look

Referee Report

This paper addresses an important and timely question concerning the relationship between longitudinal employment patterns, particularly non-standard work arrangements, and parental health across different countries. The study employs a rigorous methodological approach, using sequence analysis and regression models to examine the associations. The cross-country comparative perspective is a significant strength, allowing for insights into how different social and policy contexts may moderate these relationships.

Strengths:

• Relevance: The topic is highly relevant, given the increasing prevalence of non-standard work schedules and their potential impact on families.

• Cross-Country Design: The use of data from four developed countries (Australia, Germany, UK, and US) allows for valuable comparisons and insights into the role of different national contexts.

• Methodological Rigor: The combination of sequence analysis and regression models provides a robust approach to analyzing longitudinal work patterns and their health consequences.

• Clear Research Questions: The study is guided by well-defined research questions that are clearly stated in the introduction.

• Comprehensive Data Availability: The data is fully available without restriction, making this study highly reproducible.

Areas for Improvement:

• Sample Sizes: As acknowledged by the authors, the sample sizes for Germany are relatively small, which may limit the power to detect statistically significant effects.

• Voluntary vs. Involuntary Work Schedules: The analysis does not differentiate between voluntary and involuntary non-standard work schedules.

• Social Determinants Discussion: Could be expanded on how social determinants influence health.

• Data Comparability: Although the authors do take steps to ensure data comparability, they can be further improved.

Specific Comments and Suggestions:

• Introduction: To strengthen the introduction, the authors could briefly discuss the specific parental leave and childcare policies in the four countries under study, highlighting how they might relate to non-standard work arrangements.

• Data and Methods: Justify the age range chosen. Explain how missing data was handled and if there were any limitations with the methodology.

• Results: Provide more specific examples of differences in health outcomes due to specific reasons.

• Discussion: Focus on why parental work may carry long-lasting impacts on health, and how to deal with this issue.

Potential Typos and Misspellings:

• Page 7: "Jianghong Li, WZB Social Science Center Berlin, Germany; Curtin University and The Kids Institue, Western Australia" - "Institue" should be "Institute"

• Page 11: "sync of biological rthyms" - "rthyms" should be "rhythms"

• Page 14: "Perry-Jenkins et al" it is not consistent, some cases the last name is not hyphened. The name shoudl be the same.

• Page 15: "Specifically, the four countries examined in this study differ signifcantly in key" - "signifcantly" should be "significantly"

• Page 19: "Where the concept applies to the contry context" - "contry" should be "country"

• Page 21: "which with transforming one sequence into another"- It is "With" or "which"?

Reviewer #2: Longitudinal Employment Patterns and Parental Health: A Cross-Country Look

Summary

This paper studies how long-term employment trajectories are associated with parental physical and mental health using a life-course and cross-country perspective. Motivated by the growth of nonstandard and unstable work arrangements and cross-national differences in labor-market flexibility and health-care systems, the authors ask whether parental work schedule patterns across adulthood are linked to later health, and whether these associations vary across institutional contexts.

Using longitudinal data from Australia, Germany, the UK, and the US, the authors first apply sequence analysis to classify work schedule trajectories between ages 25 and 54. They then relate these trajectories to health outcomes measured at ages 35/40, 45/50, and 55/60 using multivariate regression models with rich sociodemographic controls.

The results show that persistent non-employment and volatile or nonstandard work trajectories are associated with worse physical and mental health across all four countries. While adverse associations are more persistent in countries with lower labor-market flexibility and weaker health-care protection (notably the US), effect sizes are larger at later ages in countries with more generous social and health systems, such as Germany and the UK.

The paper contributes by providing a comparative life-course analysis of parental work and health, highlighting how employment flexibility and institutional health-care contexts shape the long-term health consequences of unstable work arrangements.

Review

While this paper addresses a timely topic, uses rich longitudinal data, and applies a careful empirical strategy, I am left with concerns regarding (1) the motivation and relevance of the research question, particularly the implications of parental wellbeing; (2) the originality and contribution of the study, given its purely associational nature and the limited engagement with existing causal literature on work arrangements and health. Below I provide a set of comments and suggestions that I hope will help the authors clarify these issues and better articulate the paper’s contribution.

Relevance

While the paper addresses an important and timely topic, its broader relevance could be strengthened by more clearly articulating why parental wellbeing matters beyond individual health outcomes. The analysis convincingly documents associations between long-term work trajectories and parental health, but it stops short of situating these findings within a broader discussion of economic efficiency and social costs. In particular, the paper does not fully explore how deteriorations in parental health may affect labor supply, productivity, public health expenditures, or the intergenerational transmission of disadvantage. As a result, the relevance remains primarily descriptive, with the economic implications of the findings left largely implicit rather than central to the argument.

Method and Interpretation of Results

The authors are transparent about the subjective and structural aspects of the sequence analysis, including clustering decisions, treatment of missing states, and cross-country comparability constraints. These limitations are inherent to life-course sequence analysis using international panel data and are handled using standard, defensible practices, though the following points would benefit from revision:

Causality: The paper appropriately frames its results as associations rather than causal effects; however, given the academic and policy relevance of work schedules and labor-market arrangements, it is crucial to clarify these limitations from the outset. I encourage the authors to expand the discussion of causality by explicitly addressing: (i) why causal estimates would be particularly valuable in this context; (ii) what the reliance on associations implies for academic contribution and policy relevance; and (iii) how the life-course, trajectory-based approach, while not resolving endogeneity, helps mitigate some limitations of associational evidence by capturing cumulative exposure rather than point-in-time correlations.

Institutional context and cross-country interpretation: More detail on the legal and institutional context shaping labor-market flexibility and health protection across countries is needed to strengthen the interpretation of cross-country differences by more systematically linking observed patterns to institutional features such as working-time regulations, employment protection, schedule flexibility rights, and health-care system coverage.

Origininality

The paper provides a solid discussion of the literature documenting associations between work instability and parental wellbeing; however, it is not clear how by providing evidence on associations the paper contribute academically and may have implication for policy. In particular, the reasons for the existing gap in cross-country evidence sufficiently discussed, whether due to data limitations or the complexity of comparing highly heterogeneous institutional contexts. Moreover, given that the paper does not report causal estimates, it would be useful to clarify how the analysis nevertheless adds value to the literature and complements the existing causal evidence. More specifically:

• Clarify why US-centric evidence is a limitation. More explicitly articulate why the dominance of US-based studies poses an external validity problem, particularly given differences in labor-market regulation, social protection, and health-care systems across countries.

• Explain the source of the cross-country evidence gap:. Clarify whether the lack of cross-country analyses in the literature primarily reflects data constraints or whether contextual heterogeneity has limited meaningful comparison, and explain how the paper navigates these challenges and add to the literature by dealing with data of countries with such differential features.

• Position the contribution relative to existing causal evidence. There is a growing body of work providing causal estimates of the effects of work instability/flexibility or even parental leave on parental and infant wellbeing that is not fully discussed or identified in the current version of the paper. The contribution would be strengthened by acknowledging this literature and clarifying which aspects remain missing in these studies (e.g., cross-country evidence or life-course perspective) and how the present analysis complements rather than replicates existing causal findings. See some of them:

o Persson, P., & Rossin-Slater, M. (2024). When dad can stay home: fathers' workplace flexibility and maternal health. American Economic Journal: Applied Economics, 16(4), 186-219.

o Bloom, N., Liang, J., Roberts, J., & Ying, Z. J. (2015). Does working from home work? Evidence from a Chinese experiment. The Quarterly journal of economics, 130(1), 165-218.

o Chuard, C. (2023). Negative effects of long parental leave on maternal health: Evidence from a substantial policy change in Austria. Journal of Health Economics, 88, 102726.

Other comments about writing

Introduction and motivation: The Introduction does not sufficiently clarify the motivation and contribution of the paper. It would benefit from a clearer articulation of the underlying theoretical mechanisms, a precise research question, and a concise explanation of how the paper answers this question and contributes to the existing literature. As currently written, the framing feels incomplete and makes it difficult to assess the paper’s value early on.

Presentation of results: The results are largely presented as a sequence of estimates with little interpretation at the point of reporting. This gives the impression that the paper is primarily reporting numbers, with substantive discussion postponed to later sections. Integrating interpretation alongside the presentation of results is necessary to help readers understand the relevance and implications of the findings.

6. PLOS authors have the option to publish the peer review history of their article (what does this mean?). If published, this will include your full peer review and any attached files.

Reviewer #1: No

Reviewer #2: No

You may also use PLOS’s free figure tool, NAAS, to help you prepare publication quality figures: https://journals.plos.org/plosone/s/figures#loc-tools-for-figure-preparation

---

## [Author Response · Author response to Decision Letter 1]

2 Mar 2026

PONE-D-25-46936: Longitudinal Employment Patterns and Parental Health: A Cross-Country Look

RESPONSE TO THE EDITOR:

We thank the reviewers for their very helpful and constructive comments in strengthening this manuscript. We have revised the paper based on the comments raised by the reviewers and have responded to each point in the responses below. The major changes are in the tracked changes.

RESPONSE to Editorial Comment on “Journal Requirement”

Journal Requirements:

[RESPONSE] Thank you for your careful checking of our manuscript. We have carefully followed the guidelines you provided to prepare our revised manuscript.

[RESPONSE] We provide the web links to individual data in our S1. File for accessing each of the four datasets we used. We have also provided statements about data from Australia, Germany, and the UK that require an individual researcher’s application for public data access, which we have obtained. For this reason, we do not upload the data for public access. To enhance transparency in our data analysis, we have provided detailed information on how we coded each variable across all four datasets shown in S1. Table.

3. Please include a copy of Table 1 and 2 which you refer to in your text on page 16 and 20.

[RESPONSE] We have now included the files for Tables 1 and 2 in our resubmission.

[RESPONSE] We now provide the captions of our supporting information at the end of the manuscript and ensure the reference to the supporting information in the text is accurately stated.

5. We note that there is identifying data in the Supporting Information file “S1 Table_PLOS supporting information.docx, S3 Table_PLOS supporting information.docx and S4 Table_PLOS supporting information.docx”. Due to the inclusion of these potentially identifying data, we have removed this file from your file inventory. Prior to sharing human research participant data, authors should consult with an ethics committee to ensure data are shared in accordance with participant consent and all applicable local laws

[RESPONSE] We have carefully examined each table to ensure no identifiable data is presented. We used publicly available data in aggregate form (e.g., means, large counts, regression results) to present those statistics. They are not identifiable by any means. We inquired about this issue with the editorial office and received a response confirming that our tables do not contain identifiable information (response dated February 4, 2026, from Kit Stokes).

RESPONSE TO REVIEWER 1

We thank this reviewer for the thoughtful comments and believe that the paper has been considerably improved. All comments are addressed below.

Areas for Improvement:

• Sample Sizes: As acknowledged by the authors, the sample sizes for Germany are relatively small, which may limit the power to detect statistically significant effects.

[RESPONSE] We thank this reviewer for this important point. We now further clarify in Lines 639-644 that the limited sample size in Germany is confined to respondents aged 35 and its associated limitations, whereas for the other two age groups (45 and 55), the sample sizes are larger and comparable to those in Australia and the United Kingdom.

• Voluntary vs. Involuntary Work Schedules: The analysis does not differentiate between voluntary and involuntary non-standard work schedules.

[RESPONSE] We thank this reviewer for raising this important issue. We discuss the implications of (in)voluntary work schedules for parental health in our Literature Review section (Lines 215-225), and we address this limitation in Lines 662-667, given the data at hand and its implications for our results.

• Social Determinants Discussion: Could be expanded on how social determinants influence health.

[RESPONSE] As suggested, we now add a discussion directly linking SDOH to health (Lines 101-104) and specify multiple aspects of nonstandard work (including hours, instability, volatility, and structural inequality) that closely align with the SDOH concept (Lines 104-111).

• Data Comparability: Although the authors do take steps to ensure data comparability, they can be further improved.

[RESPONSE] We thank the reviewer for this important point and we wholeheartedly agree with this point. We tried to harmonize as much as possible across the datasets; however, some variables were measured differently across countries, and we thus acknowledged this in the limitations section (Lines 644-651). In addition to the descriptions provided in the text (Lines 323-349), we address the data comparability issue by conducting separate regression analyses for each country rather than pooling the datasets. This analysis allows us to examine the relationship between work patterns and health outcomes within each country rather than across countries, thereby at least partially avoiding potential measurement noise from cross-country data harmonization. We now address this issue in our Limitations section (Lines 644-658).

Specific Comments and Suggestions:

• Introduction: To strengthen the introduction, the authors could briefly discuss the specific parental leave and childcare policies in the four countries under study, highlighting how they might relate to non-standard work arrangements.

[RESPONSE] We thank this reviewer for this helpful comment to strengthen the context introduction for this paper. We now add a statement to acknowledge the important role of institutional context including childcare policies and labor market policies in the Introduction section (Lines 44-56) and provide a detailed discussion concerning institutional context in the Literature Review section (Lines 130-189), with the intention to present the information in one place without repetition. We hope you agree with our organization. Given parental leave policy does not immediately or directly relate to our key independent variable, work schedules, we focus on childcare policy because the extant research has shown how the availability and affordability of childcare and the work schedule arrangement closely shape each other; we discuss this in the Literature Review section (Lines 215-242).

• Data and Methods: Justify the age range chosen. Explain how missing data was handled and if there were any limitations with the methodology.

[RESPONSE] We now provide details on the reasons for choosing the specific age range (Lines 324-328). In addition, we now specify the details of missing cases for each variable in our supporting information (S3 Table was unfortunately removed by the editorial office in the first submission), along with how we handled missing data for the independent and dependent variables (Lines 374-382).

• Results: Provide more specific examples of differences in health outcomes due to specific reasons.

[RESPONSE] We thank this reviewer for this helpful suggestion to make the results more relatable to readers. As suggested by Reviewer 2 as well, we now include discussions to qualify some of the results from Table 2, shown in Lines 462-477 and 548-566.

• Discussion: Focus on why parental work may carry long-lasting impacts on health, and how to deal with this issue.

[RESPONSE] We thank this reviewer for this valuable point. In addition to our discussion throughout the Literature Review section, which underscores the critical implications of parental work for the well-being of the next generation, we now add a discussion to underscore the critical roles of parental work and health when paired with important policies in achieving positive child outcomes, as shown in Lines 702-709.

Potential Typos and Misspellings:

• Page 7: "Jianghong Li, WZB Social Science Center Berlin, Germany; Curtin University and The Kids Institue, Western Australia" - "Institue" should be "Institute"

• Page 11: "sync of biological rthyms" - "rthyms" should be "rhythms"

• Page 14: "Perry-Jenkins et al" it is not consistent, some cases the last name is not hyphened. The name shoudl be the same.

• Page 15: "Specifically, the four countries examined in this study differ signifcantly in key" - "signifcantly" should be "significantly"

• Page 19: "Where the concept applies to the contry context" - "contry" should be "country"

• Page 21: "which with transforming one sequence into another"- It is "With" or "which"?

[RESPONSE] We thank this reviewer for this careful reading, our sincere apologies for our carelessness in spelling. We have now corrected these typos and carefully proofread our manuscript to ensure it is free of errors.

RESPONSE TO REVIEWER 2

We appreciate the reviewer's important and valuable comments and have addressed them as follows.

Relevance

While the paper addresses an important and timely topic, its broader relevance could be strengthened by more clearly articulating why parental wellbeing matters beyond individual health outcomes. The analysis convincingly documents associations between long-term work trajectories and parental health, but it stops short of situating these findings within a broader discussion of economic efficiency and social costs. In particular, the paper does not fully explore how deteriorations in parental health may affect labor supply, productivity, public health expenditures, or the intergenerational transmission of disadvantage. As a result, the relevance remains primarily descriptive, with the economic implications of the findings left largely implicit rather than central to the argument.

[RESPONSE] We thank this reviewer for this valuable feedback to strengthen the relevance of this empirical evidence to large scholarship. We now add a discussion in the Introduction section highlighting the links between work schedules and workers’ health, the economic and productivity costs to employers and society (Lines 21-28), and the significant implications for the intergenerational transmission of disadvantages (Lines 28-36). As our aim is to underscore the importance of parental work in transmitting (dis)advantages, we further situate parental health outcomes in the context of intergenerational transmission of disadvantage, as shown throughout our literature review and in Lines 702-709, as suggested by Reviewer 1.

In addition, to anchor our focus on cross-country analysis as well as addressing the important comments on institutional context from both you and Reviewer 1, we now place interpretations and discussions aligning more with the differences in institutional contexts concerning labor market and social protection provisions (e.g., working time regulation, childcare policy) throughout the paper, we hope you agree with our approach.

Method and Interpretation of Results

The authors are transparent about the subjective and structural aspects of the sequence analysis, including clustering decisions, treatment of missing states, and cross-country comparability constraints. These limitations are inherent to life-course sequence analysis using international panel data and are handled using standard, defensible practices, though the following points would benefit from revision:

Causality: The paper appropriately frames its results as associations rather than causal effects; however, given the academic and policy relevance of work schedules and labor-market arrangements, it is crucial to clarify these limitations from the outset. I encourage the authors to expand the discussion of causality by explicitly addressing: (i) why causal estimates would be particularly valuable in this context; (ii) what the reliance on associations implies for academic contribution and policy relevance; and (iii) how the life-course, trajectory-based approach, while not resolving endogeneity, helps mitigate some limitations of associational evidence by capturing cumulative exposure rather than point-in-time correlations.

[RESPONSE] We thank this reviewer for this helpful comment. We now add discussions in the front section of the paper (Lines 57-71) to specifically acknowledge (1) the importance of causality for policy-making, (2) this study’s inability to establish causality due to data at hand and ethical concerns, and (3) the advantage of longitudinal data together with sequence analysis in capturing long-term work patterns to mitigate measurement noise and address temporal sequence, and hence the importance of accumulating empirical evidence with better and sophisticated tools when experimental design is impossible/unethical.

Institutional context and cross-country interpretation: More detail on the legal and institutional context shaping labor-market flexibility and health protection across countries is needed to strengthen the interpretation of cross-country differences by more systematically linking observed patterns to institutional features such as working-time regulations, employment protection, schedule flexibility rights, and health-care system coverage.

[RESPONSE] As suggested, we now add discussions related to institutional contexts throughout our paper in the front section of the paper (Lines 44-56), the Literature Review section (Lines 130-189), the Results section (Lines 462-477, 548-566), and the Discussion section (Lines 616-636).

Origininality

The paper provides a solid discussion of the literature documenting associations between work instability and parental wellbeing; however, it is not clear how by providing evidence on associations the paper contribute academically and may have implication for policy. In particular, the reasons for the existing gap in cross-country evidence sufficiently discussed, whether due to data limitations or the complexity of comparing highly heterogeneous institutional contexts. Moreover, given that the paper does not report causal estimates, it would be useful to clarify how the analysis nevertheless adds value to the literature and complements the existing causal evidence. More specifically:

• Clarify why US-centric evidence is a limitation. More explicitly articulate why the dominance of US-based studies poses an external validity problem, particularly given differences in labor-market regulation, social protection, and health-care systems across countries.

[RESPONSE] We thank this reviewer for these valuable points regarding the originality and contribution of this association study. We now provide a more detailed discussion of the state of knowledge on the links between parental work schedules and parental health around the globe (Lines 252-266), followed by a discussion of the potential reasons for mixed findings shown in the extant knowledge (Lines 266-282) including why one country’s findings may not be applicable to other countries (Lines 278-286).

• Explain the source of the cross-country evidence gap: Clarify whether the lack of cross-country analyses in the literature primarily reflects data constraints or whether contextual heterogeneity has limited meaningful comparison, and explain how the paper navigates these challenges

---

## [Decision Letter · Decision Letter 1]

20 May 2026

Longitudinal Employment Patterns and Parental Health: A Cross-Country Look

PONE-D-25-46936R1

Dear Dr. Han,

We’re pleased to inform you that your manuscript has been judged scientifically suitable for publication and will be formally accepted for publication once it meets all outstanding technical requirements.

Kind regards,

José Alberto Molina

Academic Editor

PLOS One

Additional Editor Comments (optional):

Reviewers' comments:

Reviewer's Responses to Questions

**Comments to the Author**

1. If the authors have adequately addressed your comments raised in a previous round of review and you feel that this manuscript is now acceptable for publication, you may indicate that here to bypass the “Comments to the Author” section, enter your conflict of interest statement in the “Confidential to Editor” section, and submit your "Accept" recommendation.

Reviewer #1: All comments have been addressed

2. Is the manuscript technically sound, and do the data support the conclusions?

Reviewer #1: (No Response)

3. Has the statistical analysis been performed appropriately and rigorously? 

Reviewer #1: Yes

4. Have the authors made all data underlying the findings in their manuscript fully available?

Reviewer #1: Yes

5. Is the manuscript presented in an intelligible fashion and written in standard English?

Reviewer #1: Yes

6. Review Comments to the Author

Reviewer #1: Thank you for the concise responses to the concerns raised. I am satisfied with the answers provided and the changes made to address the issues.

7. PLOS authors have the option to publish the peer review history of their article (what does this mean?). If published, this will include your full peer review and any attached files.

Reviewer #1: No

---

## [Editor Report · Acceptance letter]

PONE-D-25-46936R1

PLOS One

Dear Dr. Han,

I'm pleased to inform you that your manuscript has been deemed suitable for publication in PLOS One. Congratulations! Your manuscript is now being handed over to our production team.

Kind regards,

on behalf of

Professor José Alberto Molina

Academic Editor

PLOS One